



# Technical note: Improving the European air quality forecast of Copernicus Atmosphere Monitoring Service using machine learning techniques

Jean-Maxime Bertrand[1], Frédérik Meleux[1], Anthony Ung[1], Gaël Descombes[1], Augustin Colette[1]

[1] Institut National de l'Environnement Industriel et des Risques (INERIS), Parc Alata, BP2, 60550 Verneuil-en-Halatte, France

*Correspondence to*: Frédérik Meleux (frederik.meleux@ineris.fr)

**Abstract.**

Model Output Statistics (MOS) approaches relying on machine learning algorithms were applied to downscale regional air

quality forecasts produced by CAMS (Copernicus Atmosphere Monitoring Service) at hundreds of monitoring sites across Europe. Besides the CAMS forecast, the predictors in the MOS typically include meteorological variables but also ancillary data. We explored first a "local" approach where specific models are trained at each site. An alternative "global" approach where a single model is trained with data from the whole geographical domain was also investigated. In both cases, local predictors are used for a given station in predictive mode. Because of its global nature, the latter approach can capture a variety

of meteorological situation within a very short training period and is thereby more suited to cope with operational constraints in relation with the training of the MOS (frequent upgrades of the modelling system, addition of new monitoring sites). Both approaches have been implemented using a variety of machine learning algorithms: random forest, gradient boosting, standard and regularized multi-linear models. The quality of the MOS predictions is evaluated in this work for four key pollutants, namely particulate matter $PM_{10}$ and $PM_{2.5}$, ozone $O_3$ and nitrogen dioxide $NO_2$, according to scores based on the predictive

errors and on the detection of pollution peaks (exceedances of the regulatory thresholds). Both the local and the global approaches significantly improve the performances of the raw Ensemble forecast. The most important result of this study is that the global approach competes with and can even outperform the local approach in some cases. This global approach gives the best RMSE scores when relying on a random forest model, for the prediction of daily mean, daily max and hourly concentrations. By contrast, it is the gradient boosting model which is better suited for the detection of exceedances of the

European Union regulated threshold values for $O_3$ and $PM_{10}$.




## 1 Introduction

Outdoor air pollution induced by natural sources and human activities remains a major environmental and health issue
worldwide. Producing reliable short-term forecasts of pollutants concentrations is a key challenge to support national
authorities in their duties regarding the European Air Quality Directive, like planning and communications about the air quality
status toward the general public in order to limit the exposure of populations. Progress in computing technologies during the
last decades has allowed the rise of large-scale chemistry transport models (CTMs) which provide a comprehensive view of
the air quality on a given time period and geographical domain by solving the differential equations that govern the transport
and transformation of pollutants in the atmosphere. An overview of such deterministic air quality forecasting systems operating
in Europe was provided by Zhang et al. (2012). Ensembles of several CTMs models have also been used in order to improve
single model forecasts (Delle Monache and Stull, 2003; Wilczak et al., 2006). Such an Ensemble approach is currently used
in the frame of the Copernicus Atmospheric Monitoring Service (Marécal et al. 2015) to provide daily air quality forecasts
over the European territory (https://atmosphere.copernicus.eu/air-quality).

Statistical post-processing offers a way to improve the raw outputs of deterministic models, not undermining inherent
capacities of CTMs. For instance, one must acknowledge that regional scale CTMs are primarily designed to capture
background air pollution so that spatial representativeness remains a concern in the immediate vicinity of large emission
sources. Spatial downscaling is therefore a good example of the relevance of hybrid statistical and deterministic modelling,
but there are also other applications such as exceedances modelling of extreme values or even compensation of systematic
biases. Modellers are working to solve such issues by continuously improving models and input data, but post-processing
offers a pragmatic solution that must be considered.

Running-mean bias correction, Kalman-filter, and analogs (Delle Monache et al., 2006; Kang et al., 2008; Djalalova et al.,
2015) are the most widespread examples of Model Output Statistics (MOS) proposed in the literature to improve air quality
forecasts. Another very common type of MOS, is multi-linear regression statistical modelling to predict a corrected
concentration at a given location using any available information, including the deterministic forecast, meteorological
variables, or any other ancillary data. Such regression-based MOS approaches have been implemented in Europe in several
national air quality forecasting service, sometimes for more than a decade such as in the French operational forecasting system
PREV'AIR (Honoré et al., 2008; Rouïl et al; 2009). More recently, Petetin et al. (2022), performed a systematic evaluation of
one of the most exhaustive selections of MOS techniques (including Kalman Filter, Analogs in addition to hierarchical machine
learning algorithms) for the specific case of ozone forecasts in the Iberian Peninsula.

The goal of this work is to explore the use of several machine learning algorithms to improve the air quality forecasts of the
CAMS Regional Ensemble model at hundreds of monitoring sites across Europe for the ozone ($O_3$) particulate matter ($PM_{10}$



and PM$_{2.5}$) and nitrogen dioxide (NO$_2$) pollutants. When classical MOS approach is applied at each observation site, long training periods based on model outputs and observations are required. This issue is particularly pregnant in our context since,

as a regularly maintained operational model, the CAMS Ensemble model (composed of 7 members during the period of study) is subject to frequent upgrades. Therefore, an alternative "global" approach which consists in regularly training a new single model for the whole set of monitoring sites with the most recent data (a few days) has been tested for comparison. In the following article, we present first, in section 2, the observations and model data sets used to train and test the predictive models. Then, MOS approaches, and algorithms are presented in section 3. Finally, section 4 explores the sensitivity of the 2 MOS

approaches to training data and section 5 compares and discuss their performances in the frame of the selected scenarios.

**2 Training data**

The MOS development is based on three years of air pollution and meteorological data covering the 2017-2019 period. This data includes hourly in situ observations of PM$_{10}$, PM$_{2.5}$, NO$_2$ and O$_3$ concentrations at hundreds of urban, suburban, and rural background regulatory monitoring stations and is retrieved from the Up-To-Date (UTD) dataset of the Air Quality E-reporting

database (https://www.eea.europa.eu/data-and-maps/data/aqereporting-9) of the European Environment Agency. Daily mean, daily 1h maximum and daily 1h minimum where calculated when 75% of the hourly data was available for the considered dates (i.e., at least 18h over 24h). All the stations located into the European region, over a domain ranging from -25° W to 45° E longitude and 30° S to 70° N latitude have been considered in this work. The total number of stations available for training and testing the MOS is 1535 for O3, 957 for PM10, 1468 for NO2 and 498 for PM2.5.

Hourly concentrations from the CAMS European Ensemble forecast have been retrieved from the Atmosphere Data Store (https://ads.atmosphere.copernicus.eu/cdsapp#!/home). During the 2017-2019 period[1], the CAMS Ensemble was defined as the median of 7 individual models covering the European region at the resolution of 0.1° and developed by several European modelling teams, namely: CHIMERE (INERIS, France), EMEP (MET Norway, Norway), EURAD-IM (RIU-UK, Germany), LOTOS-EUROS (KNMI-TNO, The Netherlands), MATCH (SMHI, Sweden), MOCAGE (METEO-FRANCE, France), and

SILAM (FMI, Finland). Note that the CAMS Ensemble was upgraded during the month of June 2019 with the use of a new anthropogenic emissions dataset, extension of the geographical domain and addition of dust and secondary inorganic aerosols in near real time production. However, the evaluation of the MOS over the 2019 testing period was not strongly impacted by this change in the set-up since the scores remains stable before and after this upgrade. Hourly surface meteorological data was interpolated from the IFS (Integrated Forecasting System[2] – ECMWF). The specific list of meteorological variables is

discussed in Section 3.3. Both concentration and meteorological forecasts were extracted at the locations of monitoring station using a distance weighted average interpolation.

---

[1] Since then, 4 new models have been added to the Ensemble calculation, namely DEHM (Aarhus University, Denmark), GEMAQ (IEP-NRI, Poland), MINNI (ENEA, Italy) and MONARCH (BSC, Spain).
[2] https://www.ecmwf.int/en/research/modelling-and-prediction





### 3 Design of the MOS approaches

The MOS strategy can be called "hybrid" modelling in the sense that it uses both a deterministic forecast (here the CAMS Regional Ensemble) and other relevant predictors to produce a statistically corrected output concentration. Since calibration

and testing is made possible by the availability of both predictions and observations over a past period, in machine learning terminology it corresponds to a supervised learning problem. A predictive model is built from a training data set that consists in a series of concentration values observed in the past together with the corresponding predictors values. This model fitted with the training data will then be applied to future situations (new predictors values) to produce a statistically corrected concentration forecast. Three distinct problems have been considered in this work: prediction of daily mean, daily maximum

and hourly concentrations. The quality of the predictions is explored for the first day (D+0) or first 24 hours of the forecast in this work, but the methodologies proposed are adapted to tackle longer forecast leads.

#### 3.1 Machine learning algorithms

Five types of predictive models based on different machine learning algorithms are tested and compared to each other. Three of them belong to the family of the linear models, namely the standard, the LASSO and the ridge linear model. They are

formulated as (eq. 1):

$$y^* = \alpha_0 + \sum_{j=1}^{p} \alpha_j x_j \qquad (1)$$

Where $y^*$ denotes the predicted value for the pollutant's concentration, $\alpha_0$ is the intercept term, $x_j$ denotes a continuous variable or a dummy variable (taking values 0 or 1) that indicates the absence or presence of some categorical effect, $\alpha_j$ are the coefficients of the statistical model that have to be determined and $p$ the number of predictors. The coefficients are chosen

to minimize the Penalized Residual Sum of Squares (eq. 2):

$$PRSS = \sum_{i=1}^{N} \left( y_i - \alpha_0 - \sum_{j=1}^{p} \alpha_j x_{ij} \right)^2 + \lambda \sum_{j=1}^{p} f(\alpha_j) \qquad (2)$$

Where $y_i$ denotes an observed concentration and $x_{ij}$ the associated value for the predictor $j$. $N$ is the number of observations in the training data set, $\lambda$ a penalty coefficient, and $f$ denotes either the absolute-value or the square function. In the case where $\lambda$ is set to zero, the regularization term on the rightmost part of the equation nullifies and we obtain a standard linear model

(LM) based on the minimization of residual sum of squares. Otherwise, $\lambda$ will have to be tuned (see below) and depending on the choice of $f$ - absolute-value or square function - we obtain a LASSO (Least Absolute Shrinkage and Selection Operator) or a ridge linear model respectively. The ridge and the LASSO regression were introduced separately by Hoerl and Kennard (1970) and Tibshirani (1996) respectively. For both the ridge and LASSO approaches, the regularization term in eq. 2 favours solutions with coefficient values of small amplitude, thus reducing the risk of overfitting, i.e. of producing a model that stick





120 too close to the training data and has poor generalization skills. In this study, we used the implementation of the ridge and LASSO regression in the "glmnet" package in the R language (Friedman et al. 2010).

The other 2 predictive models are based on the decision trees described by Breiman et al. (1984). These trees are based on series of nodes that represent both a predictor and an associated threshold value. Each node is divided into 2 subsequent nodes until we reach a final node (a leaf) that gives the value of the prediction. The prediction function can also be seen as a partition

125 of the predictors space where each sub-region is associated to a constant output value. Decision trees are an interesting solution as they can capture complex non-linear interactions and internally handle the selection of relevant predictors. However, they suffer from poor generalization skills. To tackle this issue, Ensemble methods based on an aggregation of decision trees have been proposed. In this work we have tested two popular tree-based Ensemble algorithms, namely the random forest (RF) and the gradient boosting model (GBM). RF models were introduced by Breiman (2001). They rely on an aggregation of binary

130 decision trees that are built independently, using a bootstrap sample of the training data and randomly selecting subsets of candidate predictors at each node. The RF prediction is then given by the average of the trees predictions for regression problems or using majority vote for classification problems. Unlike Random Forest, GBM relies on relatively small trees that are built sequentially. After the first tree is trained, each subsequent tree is trained to predict the error left by the already trained Ensemble of trees. When the final number of trees is reach, the GBM prediction is given by the sum of the initial concentration

135 prediction and errors predicted by each tree. This mechanism, called Boosting, was first described by Freund & Schapire (1996) with the adaBoost algorithm for the prediction of a binary variable. The Gradient Boosting Machine algorithm is an adaptation, from Friedman (2001), for the prediction of quantitative variables. In this study we used the "randomForest" (Liaw & Wiener, 2002) and "gbm" (Greenwell et al., 2019) R packages for the implementation of the RF and GBM algorithms, respectively.

140 A key challenge with statistical learning methods is to learn as much as possible from the training data, without losing generalization skills. To reach an optimal balance and optimize the predictive performances, a learning algorithm may be tuned, prior to the training phase, by choosing values for some parameters often referred to as hyper-parameters. The tuning of hyper-parameters is performed at every monitoring site for the local MOS approach or every day for the global MOS approach (local and global approaches are defined in section 3.2). The method for tuning the hyper-parameters consists in a

145 grid search where possible values for each hyper-parameter are pre-defined and every possible combination is tested using a 5-fold cross validation procedure. The number of parameters to be tuned depends on the algorithm. It is limited to 1 for the LASSO and ridge model, 2 for the random forest and 4 for the GBM. To limit the number of combinations and computation time, the grids of possible values for each parameter were kept simple, with very few values to test, and remained the same in all the learning configurations of this study. The tuning of each algorithm was performed using the caret R package (Kuhn,

150 2008). The grids of tuning values for each algorithm are described in appendix A.



### 3.2 Local and global approaches

The first approach tested in this work is local, meaning that a different MOS model is built for each observation station. This approach is implemented for example in the French national forecasting system PREV'AIR (Honoré et al., 2008). As each model is trained with local data only, we expect that it will be able to correct the deterministic model output in a way that reflects local specificities contributing to the station representativeness. A limitation of this local approach is that it requires long timeseries of model output and observations (with constant model formulation and set-up over this period) to build an optimized predictive model at each observation site. Any upgrade of the modelling system that might sensitively impact the model behaviour and performances might lead to a deterioration of the MOS performances and thereby requires resource consuming for re-running simulations with a consistent set-up over past period in order to build updated MOS. Newly installed observation stations will not be integrated into the MOS until enough data is gathered to train a robust model (typically at least a full year). Moreover, this local approach is optimized if the conditions (model set up, input data) during the predictions remain closed to that of the training period. In practice this might not be the case, for example because of a drastic reduction of pollutants emissions due to local action plans or even not anticipable circumstances such as the drop in activity induced by the COVID crisis. In such situations, the local MOS correction might be biased due to inadequacy with the training period's conditions. This feature is interesting and has been exploited for example to assess the impact of COVID-19 lockdown upon NO2 pollution in Spain (Petetin et al., 2020) based on a "business as usual" concentration correction, following the meteorological normalisation method by Grange et al., 2018. However, there is also a need for more flexible MOS approaches that rapidly adapt to unanticipated changes in emissions. In the present study, the local approach was investigated using 2017 and 2018 data for training the MOS and using 2019 data to evaluate its performances.

The second approach, called "global", has been designed to address operational constraints such as the CTMs upgrades or changes in the network or observations. The idea is to build a single global model with data coming from the whole set of observation stations. Even if a single model is derived for Europe, it is subsequently used in predictive mode with local predictors for each station. Because of their spatial distribution over the European domain, a large variety of meteorological situations can be captured within a relatively small (a few days) training period. Due to the seasonal variability, a new model must be trained regularly with the most recent data in order to remain close to new forecasting situations. In this study a new global model was trained every day using the last 3 days, the last 7 days or the last 14 days as training data and was applied to predict the concentrations of the upcoming day. This process was repeated 365 times to mimic an operational system running over the 2019-year period. With this global approach, any change in the CTM formulation will automatically be echoed into the MOS within a few days (depending on the choice for the training period duration). An important shortcoming of such a global approach is to ignore the local specificity in individual MOS models, whereas one of the main benefits of MOS approaches applied in addition to CTM results is precisely to remove systematic biases, for instance induced by spatial representativeness limitation of the models. To tackle the varying spatial representativeness of the stations, the CTM raw concentration output at each station was replaced by an "unbiased concentration" predictor, meaning the raw concentration





minus the average error of the CTM at the station during the training period. As such, the global approach combines hierarchical or regression machine learning algorithm and moving average (Petetin et al., 2022) unbiasing. This strategy will for instance lead to distinct MOS predictions at 2 stations with comparable meteorological and raw concentration forecasts (e.g. 2 stations located into the same grid cell). We also expect that this approach will better adapt to rapid changes in emissions induced by the situations mentioned above (e.g. pollution mitigation policies, COVID crisis).

### 3.3 Predictors

Increasing the number of predictors might improve the performances of a model but can also lead to overfitting and poor performances if not correctly handled by the machine learning algorithm. We have carried out tests with different sets of predictors in order to evaluate the risks and benefits from adding predictors depending on the machine learning algorithm considered. The following table details the sets of predictors that have been tested. These predictors fall into four categories: Ensemble forecast, meteorological forecast, observations and other. MOS models have been trained to work both on an hourly

basis and on a daily basis (to focus on the prediction of daily means or daily max). When designing an hourly model, all the quantitative predictors are hourly means, either forecasted for the considered time horizon (Ensemble and meteorological forecasts) or observed during the previous day at the same time. When designing a daily model, a selection procedure is achieved before the training in order to choose between the daily mean, daily min and daily max of each physical quantity the one which is best correlated with the output variable.

Set1 is the base set of predictors. It includes the Ensemble forecasts for the 4 pollutants, a first selection of surface meteorological variables (namely the temperature, relative humidity, zonal and meridional wind speed and boundary layer's height), as well as observations of the previous day. The categorical day of week predictor was only used with the local approach which includes a long training period. For the global approach, tests have been performed using either the raw Ensemble forecast or the unbiased concentration of the target pollutant as a predictor. The unbiased concentration is defined

as the forecasted Ensemble concentration minus the bias observed at the station during the previous days (days of the chosen training period). Set2 includes set1 predictors plus 4 additional meteorological predictors, namely the shortwave radiation, the surface pressure, the cloud cover and precipitations.

| Set name | Ensemble Forecasts* | Meteorological Forecasts | Observations | Other |
|---|---|---|---|---|
| Set1 (base) | PM$_{10}$ O$_3$ NO$_2$ PM$_{2.5}$ | Temperature (2 m) Relative Humidity (2 m) Wind speed (10 m) Boundary layer's height | Obs. of the previous day | Day of week (7 levels) ** |
| Set2 | PM$_{10}$ | Temperature (2 m) Relative Humidity (2 m) | Obs. of the | Day of week |




|  |  |  |  |
|---|---|---|---|
| O$_3$ | Wind speed (10 m) | previous day | (7 levels) ** |
| NO$_2$ | Boundary layer's height |  |  |
| PM$_{2.5}$ | Shortwave radiation |  |  |
|  | Surface pressure |  |  |
|  | Cloud cover |  |  |
|  | Precipitations |  |  |


Table 1: Sets of predictors used in the MOS. * Raw or unbiased (for the global approach only) concentration forecasts. ** Only for the local (or long training) approach.

**4 Sensitivity of the MOS to training data and predictors**

Specific local approach simulations have been carried out to evaluate O$_3$ daily max and PM$_{10}$ daily mean predictions

performances with various input data configurations. Only O$_3$ daily max and PM$_{10}$ daily mean predictions have been considered in this preliminary analysis in order to limit the number of simulations. These forecasts being critical in Europe because of the frequent exceedances of the regulatory threshold values that determine pollution peaks (180 µg m$^{-3}$ for O$_3$ daily max and 50 µg m$^{-3}$ for PM$_{10}$ daily mean). Each pollutant was tested with 2 configurations regarding the size of the training data set. For O$_3$, one summer (June to September 2018) or two summers (June to September 2017 and 2018) have been used as training

data sets. For PM$_{10}$, year-round data have been used, either 1 year (2018) or 2 years (2017 and 2018). A training period limited to summer months has been chosen for O$_3$ to optimize the performances during this season which is regularly subject to critical concentration levels. Similarly, model could be optimized for the cold season using winter months for training and year-round modelling could be achieved switching from one model to the other at some point during the inter-season. But we chose to limit our analysis to the hot season when most pollution peaks happen. In addition, both configurations have been tested using

2 distinct sets of predictors, namely Set1, the simplest (includes the base predictors plus the categorical day of week predictor), and Set2, including four additional meteorological predictors (see table 1). Performances have been evaluated with 2019 data, over the summer season (June to September) for O$_3$ and whole year for PM$_{10}$. As expected, the RMSE of the local MOS score average over all the monitoring stations, shown in Figure 1, is significantly reduced in comparison to that of the raw Ensemble model: by construction, the MOS approaches are unbiased and therefore remove a large part of the RMSE. The use of larger

data sets is beneficial for all the machine learning algorithms tested and is particularly interesting for O$_3$ daily max predictions (RMSE strongly decreases when using 8 months of summer data instead of 4). Results also suggest being very careful with the choice of predictors, using more predictors as in Set2 generally lead to no improvement or even a loss in performances, especially if the algorithm is not designed to handle over-fitting and if the training period is too short (see the deterioration of the O$_3$ RMSE when using the larger set of predictors Set2 in the standard linear model). Regularized linear models (ridge and

LASSO) give the best RMSE scores independently of the set of predictors and of the size of the training period. With 2017 and 2018 data for training, and the simplest set of predictors (red bar), the RMSE reaches 13.0 µg m$^{-3}$ for O$_3$ daily max





(decrease of 32 % compared to the raw Ensemble) and 5.64 µg m$^{-3}$ for PM$_{10}$ daily mean (decrease of 45 %). The Pearson correlation reaches 0.86 for O$_3$ (against 0.81 for the raw Ensemble) and 0.83 for PM$_{10}$ (against 0.7 for the raw Ensemble). See Appendix B, figure 1 to 2 for the mean bias and correlation scores with the distinct local approach modelling configurations.


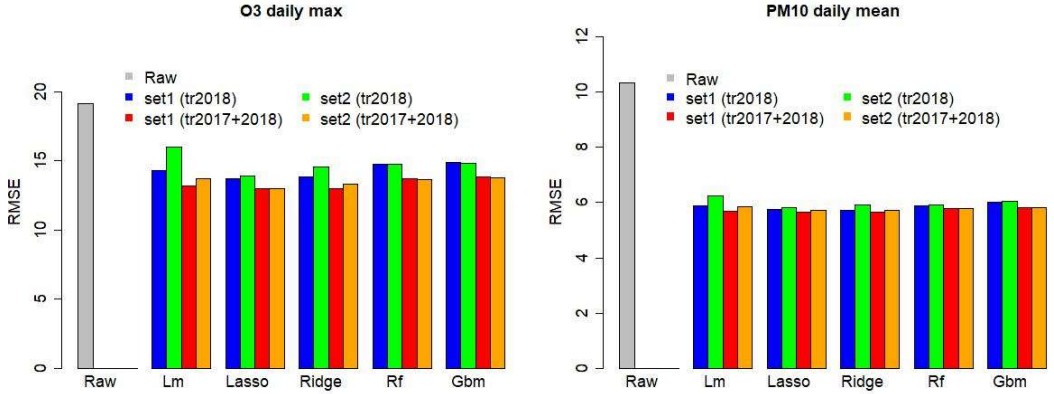

**Figure 1: RMSE score for the raw Ensemble (Raw) and local MOS approaches with the linear model (Lm), the LASSO (Lasso), the ridge (Ridge), the random forest (Rf) and the gradient boosting model (Gbm), depending on the training period and set of predictors. RMSE score is averaged over 1535 stations for O$_3$ and 957 stations for PM$_{10}$. Evaluation done over 2019 summer months for ozone**
**and whole year 2019 for PM$_{10}$.**

For the global approach, tests have been performed over the same 2019 periods (summer for O$_3$ and whole year for PM$_{10}$) with the simplest set of predictors Set1 to evaluate O$_3$ daily max and PM$_{10}$ daily mean MOS prediction according to the size of the training period (3 days, 7 days and 14 days) and the use of the raw (biased) or unbiased concentration forecasts as predictor.
Figure 2 illustrates the decrease in RMSE when using unbiased concentrations instead of raw concentrations (compare the blue and plain green bars for 3 days training). RMSE can further be improved using 7 days as training period or even 14 days for PM$_{10}$ daily mean. The random forest model gives the best RMSE scores independently of the length of the training period. With 14 days for training, and using the unbiased concentration predictor, the RMSE reaches 12.5 µg m$^{-3}$ for O$_3$ daily max (decrease of 34.6 % compared to the raw Ensemble) and 5.5 µg m$^{-3}$ for PM$_{10}$ daily mean (decrease of 46.7 %). The Pearson
correlation reaches 0.85 for O$_3$ (against 0.81 for the raw Ensemble) and 0.83 for PM$_{10}$ (against 0.7 for the raw Ensemble). See Appendix B, figure 3 to 4 for the mean bias and correlation scores with the distinct global approach modelling configurations.

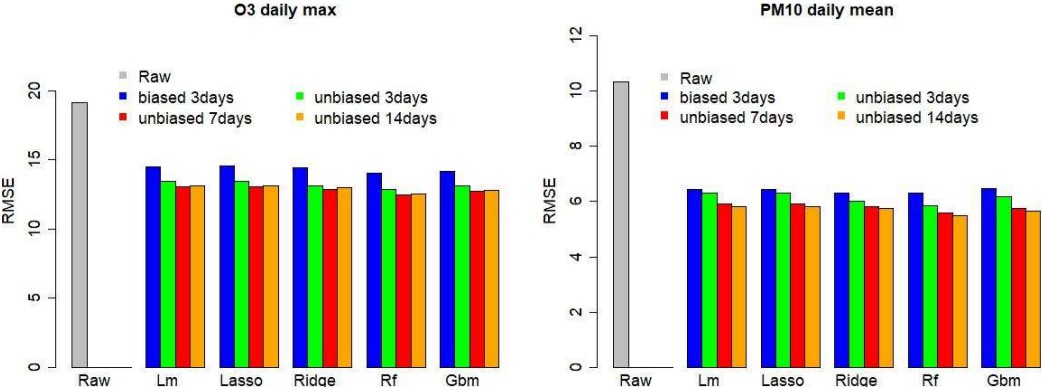


**Figure 2: RMSE score for the raw Ensemble (Raw) and local MOS approaches with the linear model (Lm), the LASSO (Lasso), the ridge (Ridge), the random forest (Rf) and the gradient boosting model (Gbm), depending on the training period and the use of biased or unbiased concentration predictors. RMSE score is averaged over 1535 stations for O₃ and 957 stations for PM₁₀. Evaluation done over 2019 summer months for ozone and whole year 2019 for PM₁₀.**

### 5 Comparison of the local and global MOS approaches

For the 4 pollutants, $O_3$, $PM_{10}$, $NO_2$ and $PM_{2.5}$, the local and global MOS have been designed for the prediction of daily mean, daily max and hourly concentrations and compared to each other. For both the local and global approaches, since the benefit of using 4 additional predictors in Set2 compared to Set1 was infirmed in Section 4, we used the simplest sets of predictors (Set1, with unbiased concentrations for the global approach). Moreover, we used in this section the more realistic scenario where only one full year of data (2018) is available for training the local approach models and 3 days for the global model. As mentioned above, performances can be optimized using larger training periods, but we chose to test the less resource consuming scenario considering operational constraints. Table 2 shows RMSE scores average over the full set of monitoring stations across Europe with the 2019 testing period. As in the previous section, evaluation is focused on the June to September period for $O_3$ and whole year for $PM_{10}$, $PM_{2.5}$ and $NO_2$.

The random Forest is particularly adapted to optimize the RMSE of the global MOS approach as the best scores are obtain with this model for the 4 pollutants and for the predictions of daily mean, daily max and hourly concentrations. Depending on the prediction objective and on the pollutant, the improvement compared to the raw Ensemble oscillates between 48.1% (decrease in RMSE) and 21.9%. The choice of the best algorithm is not that clear for the local MOS approach. Random Forest gives the best RMSE for the prediction of hourly means, but the LASSO and ridge linear models performs the best for daily means and daily max predictions. RMSE decreases oscillate between 54.1% ($NO_2$ daily max) and 20% ($PM_{2.5}$ hourly mean) with the best model scenarios, for this local approach. Still considering the best model scenarios, differences between the local





and global approach reach 6.3%, in favour of the local approach, for $NO_2$ daily max predictions and 4.6%, in favour of the global approach, for $O_3$ daily max predictions. Table 3 presents the RMSE scores for the daily mean and daily max extracted from hourly MOS predictions. These scores are comparable with those of the models specifically trained for daily mean
predictions but are significantly degraded for daily max predictions. As an example, the global approach with random forest model reduces the RMSE by 20.5% when daily max values are extracted from hourly predictions, against a reduction of 32.8% with the same model trained for daily max prediction. Therefore, depending on applications, one might consider using daily MOS instead of hourly MOS if performances must be optimized for the daily max statistics.

| | | | Local | | | | | Global | | | | |
| --- | --- | --- | --- | --- | --- | --- | --- | --- | --- | --- | --- | --- |
| | | Raw | Lm | Lasso | Ridge | Rf | Gbm | Lm | Lasso | Ridge | Rf | Gbm |
| RMSE | $O_3$ | 16.1 | 38.9% | 41.9% | 41.2% | 35.0% | 35.4% | 41.5% | 41.5% | 43.4% | 44.1% | 42.8% |
| for | $PM_{10}$ | 10.3 | 43.1% | 44.2% | 44.6% | 42.9% | 41.9% | 39.1% | 39.1% | 41.8% | 43.4% | 40.3% |
| daily | $NO_2$ | 9.6 | 53.8% | 53.9% | 54.1% | 50.3% | 51.8% | 46.0% | 46.1% | 47.5% | 48.1% | 46.9% |
| mean | $PM_{2.5}$ | 6.6 | 35.2% | 36.2% | 36.5% | 36.1% | 34.4% | 32.0% | 32.1% | 34.8% | 37.1% | 33.6% |
| RMSE | $O_3$ | 19.1 | 25.3% | 28.2% | 27.8% | 23.0% | 22.3% | 29.8% | 29.8% | 31.5% | 32.8% | 31.3% |
| for | $PM_{10}$ | 25.2 | 34.8% | 35.6% | 35.7% | 34.1% | 30.4% | 29.0% | 29.3% | 31.5% | 32.5% | 29.6% |
| daily | $NO_2$ | 21.9 | 47.9% | 48.2% | 48.4% | 46.0% | 46.0% | 39.6% | 39.7% | 41.6% | 42.1% | 40.2% |
| max | $PM_{2.5}$ | 14.9 | 31.9% | 32.9% | 32.9% | 32.2% | 29.0% | 26.6% | 27.1% | 29.6% | 30.7% | 26.7% |
| RMSE | O3 | 22.6 | 27.5% | 27.7% | 27.9% | 28.7% | 27.1% | 25.1% | 25.1% | 25.7% | 28.5% | 27.0% |
| for | $PM_{10}$ | 13.3 | 24.7% | 24.7% | 25.0% | 25.0% | 22.5% | 22.9% | 22.7% | 23.8% | 25.3% | 22.7% |
| hourly | $NO_2$ | 12.6 | 31.4% | 31.4% | 31.3% | 33.3% | 32.3% | 26.9% | 26.9% | 27.6% | 29.2% | 27.8% |
| mean | $PM_{2.5}$ | 8.6 | 19.5% | 19.6% | 19.8% | 20.0% | 17.1% | 18.6% | 18.4% | 19.8% | 21.9% | 17.8% |

Table 2: 2019 RMSE score (average of 1535 ($O_3$), 957 ($PM_{10}$), 1468 ($NO_2$) and 498 ($PM_{2.5}$) stations) for daily mean, daily maximum and hourly mean as percentage of decrease compared to the raw model RMSE. Raw model RMSE in μg m$^{-3}$ is indicated in the "Raw" column.

| | | | Local | | | | | Global | | | | |
| --- | --- | --- | --- | --- | --- | --- | --- | --- | --- | --- | --- | --- |
| | | Raw | Lm | Lasso | Ridge | Rf | Gbm | Lm | Lasso | Ridge | Rf | Gbm |
| RMSE | $O_3$ | 16.2 | 40.1% | 40.5% | 41.2% | 41.7% | 41.3% | 40.2% | 40.1% | 42.0% | 43.8% | 41.9% |
| for | $PM_{10}$ | 10.4 | 42.2% | 42.3% | 42.5% | 43.3% | 43.0% | 38.8% | 38.5% | 40.5% | 43.0% | 40.9% |
| daily | $NO_2$ | 9.5 | 52.1% | 52.1% | 51.9% | 53.6% | 54.1% | 45.5% | 45.6% | 46.5% | 47.8% | 46.9% |
| mean | $PM_{2.5}$ | 6.6 | 34.5% | 34.7% | 34.4% | 36.1% | 35.5% | 31.7% | 31.1% | 33.8% | 37.6% | 34.2% |
| RMSE | $O_3$ | 19.4 | 24.5% | 24.7% | 24.5% | 25.1% | 25.5% | 16.0% | 15.9% | 15.6% | 20.5% | 20.3% |
| for | $PM_{10}$ | 25.9 | 26.5% | 26.5% | 26.1% | 29.0% | 30.2% | 22.9% | 22.9% | 23.2% | 27.3% | 25.6% |
| daily | $NO_2$ | 22.2 | 35.7% | 35.6% | 33.9% | 39.0% | 43.1% | 32.1% | 33.0% | 32.6% | 36.1% | 36.5% |
| max | $PM_{2.5}$ | 15.2 | 24.9% | 24.5% | 24.9% | 27.6% | 28.0% | 26.6% | 27.1% | 29.6% | 30.7% | 26.7% |



Table 3: 2019 RMSE scores as percentage of decrease compared to the raw Ensemble model for the daily mean and daily max extracted from the hourly MOS predictions.


For the 4 pollutants investigated in this study, this reduction in RMSE score is associated with a strong decrease in the mean bias. As illustrated in Figure 3 for the prediction of hourly concentrations, the raw Ensemble model tends to over-estimate $O_3$ levels and to under-estimate $PM_{10}$, $PM_{2.5}$ and $NO_2$ concentrations in Central Europe (EUC), Northern Europe (EUN), Southern Europe (EUS) and Western Europe (EUW). These biases are well corrected by both the local and global MOS (see the red and
blue bars which represent the local and global MOS approaches with their respective best model scenarios). The reduction in RMSE is also associated with a significant increase in the correlation score. Similar results have been obtained with the MOS designed for daily mean and daily max (see appendix C). While the local and global approaches compete with each other for $O_3$, $PM_{10}$ and $PM_{2.5}$ daily and hourly forecasts, the local approach outperforms the global approach for the $NO_2$ pollutant. This difference is attributed to the local nature of this pollutant, i.e., the fact that concentration levels are more influenced by local
emission, and to a smaller extent by meteorological conditions. However, the global MOS approach still clearly improves performances compared to the raw Ensemble model for this pollutant.












**Figure 3: Comparison of the raw Ensemble model and best model scenarios for the local and global MOS approaches. Scores include**
**stations means of RMSE, mean bias and correlation for the prediction of hourly mean concentrations over Central Europe (EUC),**
**Northern Europe (EUN), Southern Europe (EUS) and Western Europe (EUW).**

The European Union has defined concentration thresholds to characterize pollution peaks. Exceedance of these thresholds require to inform the exposed population and the set-up of mitigation actions by local authorities to reduce the adverse effects
of the pollution. We therefore paid a special attention to the ability of the models to detect such thresholds exceedances. The threshold value of 180 $\mu g\ m^{-3}$ for $O_3$ daily max concentration and 50 $\mu g\ m^{-3}$ for $PM_{10}$ daily mean are regularly exceeded in Europe. To assess the ability of a model to detect these exceedances, we use the so-called contingency table which counts the number of good detections (predicted and observed exceedances), missed (observed but not predicted) and false alarms (predicted but not observed) over the whole set of monitoring stations. Figure 4 represent the contingency table for $O_3$ daily
max exceedances and $PM_{10}$ daily mean exceedances of the raw Ensemble model and the local MOS. The persistence model, referred to as "Pers" has been added to the plot as a reference. It is a trivial model which consists in forecasting for the oncoming day the concentration that we observed during the previous day. To characterize detection skills, 4 scores can be derived from the contingency table and plotted into a single performance diagram (Figure 5 to 8). The Probability Of Detection (on the y-axis) is defined as the ratio of good detections to the total number of observed exceedances, the Success Ratio (x-axis) is
defined as the ratio of Good Detections to the total number of predicted exceedances, the Critical Success Index (black curves) is the ratio of Good Detections to the total number of predicted or observed exceedances and the Frequency Bias (dashed straight line) is the ratio of the total number of predicted exceedances to the total number of observed exceedances. All these scores take values between 0 and 1, except for the Frequency bias which takes any positive value. A perfect model would take the value of 1 for all these scores and would be located on the upper right corner of the performance diagram. Figure 5 and 6
illustrate the performances of the local (left) and global (right) daily MOS approaches for the detection of $O_3$ and $PM_{10}$ exceedances. For $O_3$, the high value (close to 0.8) of the success ratio for the raw Ensemble model means that when it detects a threshold exceedance, there is a high probability to actually observe a threshold exceedance. But the downside is that observed exceedances have a very low probability to be detected by this model as illustrated by the very low Probability Of Detection (y-axis). In other words, the raw model is strongly biased (in frequency) with much more observed than predicted
exceedances. In contrast, the MOS allow to get a frequency bias closer to 1, reducing the success ratio but greatly improving the probability of detection. Both the local and global approaches enable to improve the overall detection performances, reaching Critical Success Index (CSI) scores between 0.3 and 0.4. The small loss in the Success Ratio is largely compensated by the gain in Probability Of Detection. In that configuration, with 4 months of data for training, the local approach works better with linear models (standard, LASSO and ridge) than with tree-based models (RF and GBM). The best CSI score is
obtained with the global approach and GBM model (0.34). This is much better than the persistence model which produces a CSI score of 0.22. Note that by construction, the Frequency Bias of the persistence model (grey circle in the performance diagrams) is equal to one (i.e. located over the bisector of the performance diagram) since the number of predicted exceedances



always equals the number of observed exceedances (exceedances are predicted with 1 day of delay). The position on the bisector line depends on the length of the episodes. Long episodes of exceedances (several consecutive days) will tend to

produce good scores (closer to the upper right corner of the performance diagram). Results are comparable for the detection of exceedances of $PM_{10}$ daily mean threshold (Figure 6), with Success Ratio scores between 0.63 and 0.68, Probability of detection between 0.42 and 0.49 and CSI between 0.35 and 0.39 depending on the MOS approach (local or global) and on the model considered. Best CSI score of 0.39 is obtained with the local approach associated with a linear model (standard, LASSO or ridge). It is interesting to note that the random forest and gradient boosting models clearly fail for the detection of $O_3$ daily

maximum threshold exceedances when used with the local approach (see the very low probability of detection Figure 5). An explanation for this behaviour is the impossibility for tree-based models to extrapolate outside the range of the original training set. For the local approach, where individual models are built at each site, this means that the prediction of threshold exceedance are only possible at sites which have been exposed to such pollution peaks during the training phase. In our context, 34 % of the sites which were exposed to $O_3$ exceedances in 2019 (testing period) were never exposed during the 2018 training period.

This ratio falls to 11 % for $PM_{10}$ threshold exceedances, which might explain the relatively correct detection skills of tree-based models for this pollutant. For the global approach, this extrapolation weakness of tree-based models might also lead to missed detections for example when pollution peaks happen after a period of several consecutive days without any threshold exceedances (considering the whole set of stations), but the impact on detection skills seems limited in this context. The detection skills of the hourly MOS are comparable to those of daily MOS for the $PM_{10}$ daily mean threshold (compare figures

6 and 8) and clearly degraded for the $O_3$ daily max threshold (compare figures 5 and 7). Nevertheless, the hourly approach enables to improve the detection skills of the raw Ensemble, with a CSI reaching 0.26 with the global approach and gradient boosting model.





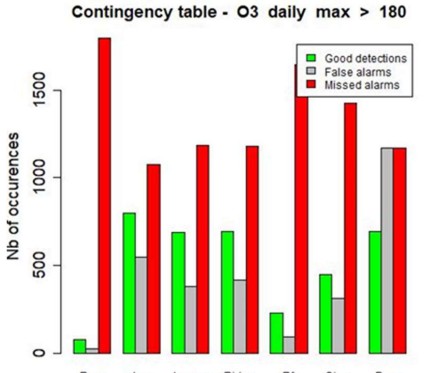 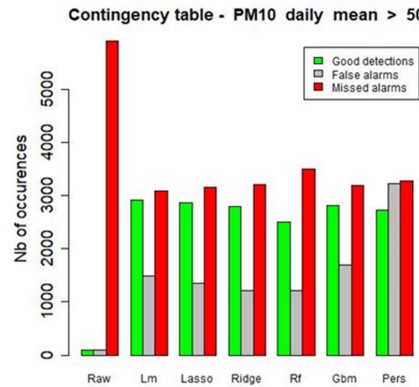

**Figure 4: Contingency table of the raw Ensemble, the local MOS models and the persistence model, over the 2019 testing period, for O₃ (left) and PM₁₀ (right) thresholds exceedances**






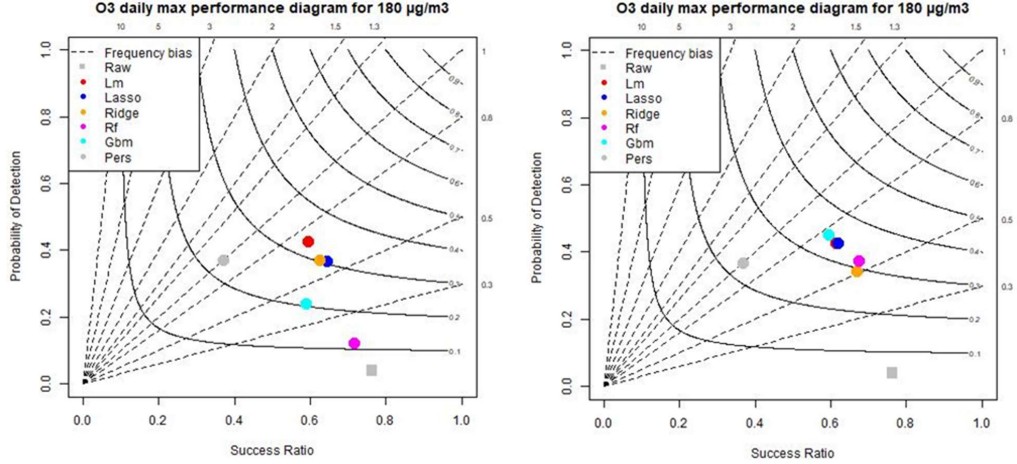


**Figure 5: Detection scores for the local (left) and global (right) daily MOS approaches for O₃ daily max 180 µg m⁻³ threshold**

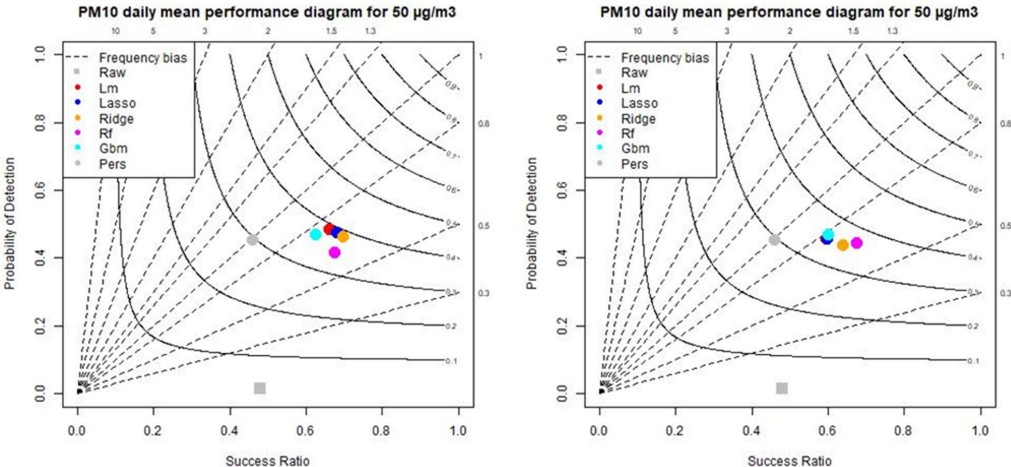

**Figure 6: Detection scores for the local (left) and global (right) daily MOS approaches for PM₁₀ daily mean 50 µg m⁻³ threshold**





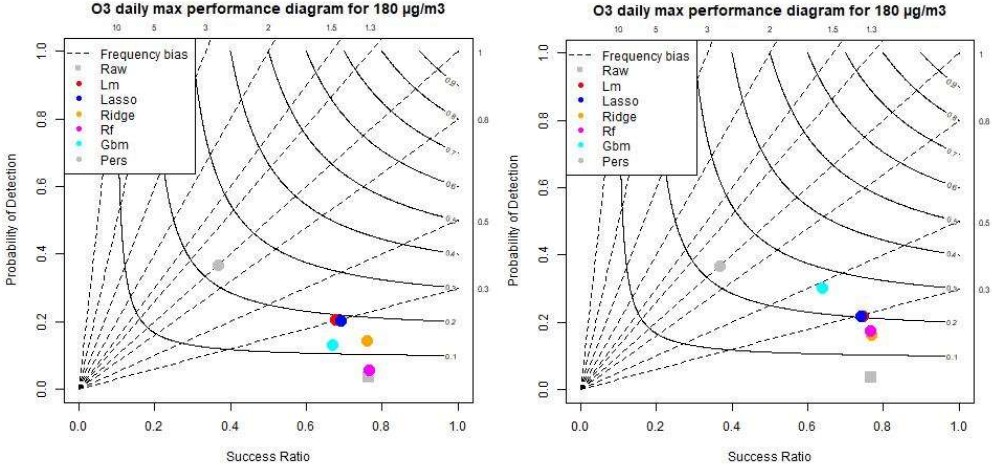

**Figure 7: Detection scores for the local (left) and global (right) hourly MOS approaches for O$_3$ daily max 180 µg m$^{-3}$ threshold**

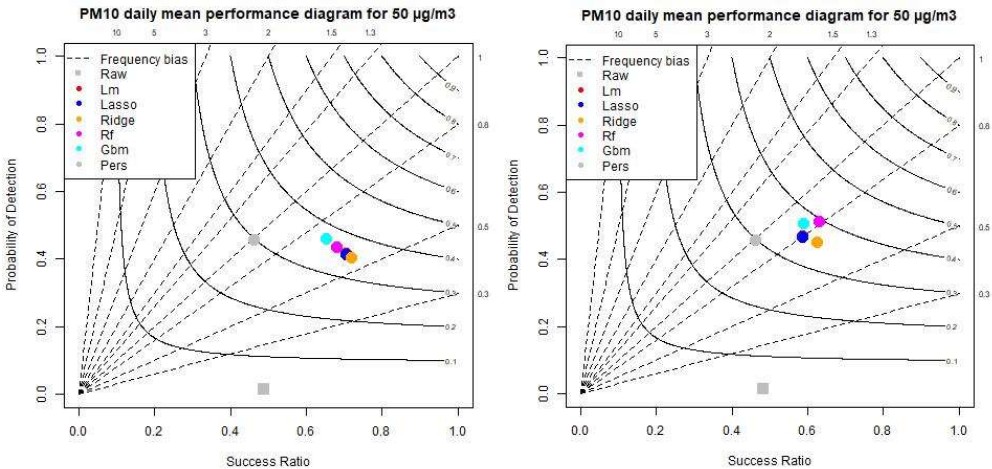

**Figure 8: Detection scores for the local (left) and global (right) hourly MOS approaches for PM$_{10}$ daily mean 50 µg m$^{-3}$ threshold**



## 6 Conclusion


This work allows to compare the performances of two MOS approaches for the correction of the CAMS forecasts at horizon D+0 for 4 regulated pollutants and at various time scales (daily and hourly), at monitoring sites covering the European territory. Both approaches (local and global) are implemented with 5 distinct machine learning algorithms ranging from simple linear regressions to more sophisticated tree-based models. The construction of optimized local MOS needs relatively large periods

of data available for training individual models at each site. It was therefore tested with a reasonable scenario, where a full year of training data was available for $PM_{10}$, $PM_{2.5}$ and $NO_2$ pollutants. For $O_3$, we focused on summer predictions and the MOS was trained with 4 months of summer data. In this context, the local MOS approach performs best with the linear models for the RMSE of daily predictions and for detection performances (CSI), while the random forest model gives the best RMSE scores for the hourly predictions. We insist that this result is only true with one year (4 summer months for $O_3$) of training

data. It could be different with shorter training period as linear models are more prone to overfitting as suggested by the results described in section 4. The global MOS is an innovative approach designed to cope with operational constraints. Its very short training period (3 days) allows to adapt in a short time to any changes in the modelling system (upgrade of the deterministic model, addition of new monitoring stations). In addition to its operational flexibility, the global approach shows performances that compete with those of the local approach. For this global approach, the random Forest algorithm gives the best RMSE

scores whatever the pollutant and time scale considered. However, if the MOS is designed for hourly prediction, the GBM algorithm is more adapted to detect $O_3$ daily max threshold exceedances. We would therefore recommend the GBM model in that situation. But one might also consider using a MOS specifically designed for daily maximum predictions to further improve detection skills.

As mentioned above, the local approach was performed in this study with relatively large training data set. Interestingly, such

a local approach was tested with CAMS $O_3$ forecasts by Petetin et al 2022, using a selection of MOS methods (including basic methods such as persistence or moving average to more sophisticated methods such as GBM) to build a model at every monitoring station located in the Iberian Peninsula. To compare the distinct MOS methods, Petetin mimics a worst-case operational scenario where very few prior data is available for training, i.e., new models are trained regularly with a growing history, starting with 30 days and ending with 2 years of data for a February 2018 to December 2019 simulation. Performances

cannot be directly compared to this work because of their distinct spatial and temporal (year-round versus summer months) coverage. Nevertheless, the authors highlight, that the GBM model present poor detection skills (worse than the persistence model) despite having the best RMSE and correlation performances. Our study confirms this result for the GBM and random forest models, even with 4 summer months for training, and we attributed these weak detection skills to the incapacity of tree-based models to extrapolate outside the range of the training data set. We demonstrated that with a constant 3-days training

period, the global approach offers stable performances, with optimized continuous and categorical skills, from the very first days following a deterministic modelling system upgrade. But in the future, such a global approach could also be used with a gradually expanding training dataset as in Petetin et al, being mindful however on the computing demand of automated learning



of such a MOS in an operational setup. Because of its flexibility, we also expect that this global approach is prone to adapt in real time to rapid changes in pollutant emissions as experimented during the COVID crisis. Further investigation could be made using 2020 data to test this approach in such a situation.


Data availability

The modelling results used in the present study

are archived by the authors and can be obtained from the corresponding author upon request


Author contribution

JMB worked on the implementation of the study and performed the simulations with support from all the co-authors. AU was responsible for the acquisition of the observed air quality data. JMB performed the analysis with the support of GD, AU, FM and AC for results interpretation. JMB wrote this article, with contributions from FM and AC.


Competing interests

The authors declare that they have no conflict of interest.

Acknowledgements

The development towards this work has received support from the Copernicus Atmosphere Monitoring Service of the European Union Implemented by ECMWF under Service Contracts CAMS_63 and CAMS2_40, the Project Lock'Air of the AQACIA/ADEME Research Project, and from the French Ministry in Charge of Ecology.







**Appendix A : Grids of tunning values for the hyper-parameters of each algorithm**

LASSO:

Lambda in {0, 0.05, 0.1 to 5.0 by increments of 0.1, 6, 7, 8, 10, 12, 15}

Ridge:

Lambda in {0, 0.05, 0.1 to 5.0 by increments of 0.1, 6, 7, 8, 10, 12, 15}

Random forest:

Number of trees = 100

Mtry = floor(sqrt(P)) largest integer less than or equal to the square root of P, where P is the number of predictors

GBM:

Interaction.depth in {2, 7}

Shrinkage in {0.05, 0.1, 0.3}

n.trees = 100

n.minobsinnode in {1, 5}

**Appendix B**




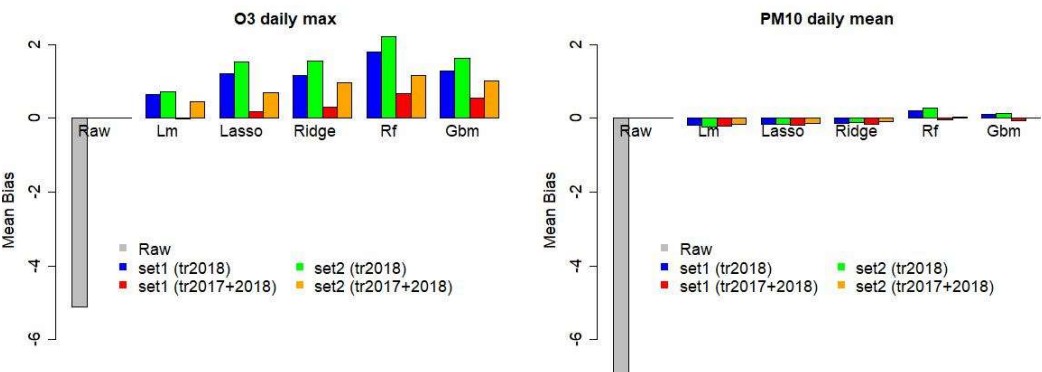

**Figure B1: Mean Bias score for the raw Ensemble model and the local MOS approach with 4 training configurations**


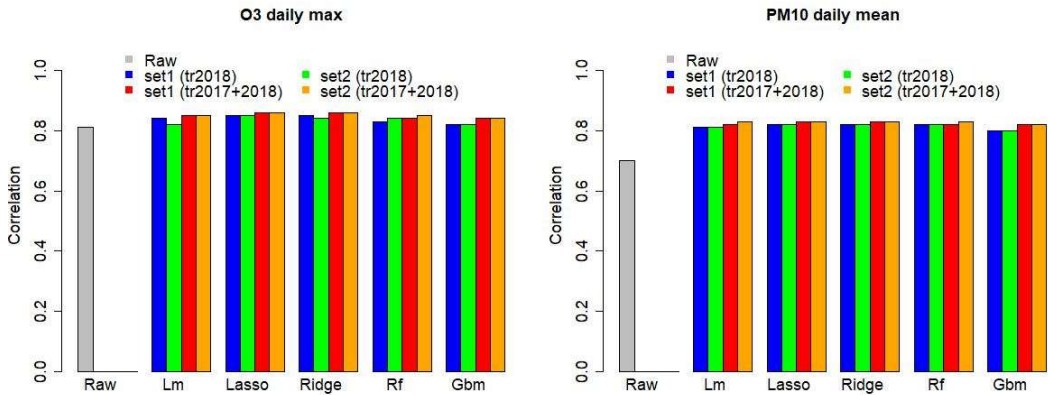

**Figure B2: Correlation score for the raw Ensemble model and the local MOS approach with 4 training configurations**





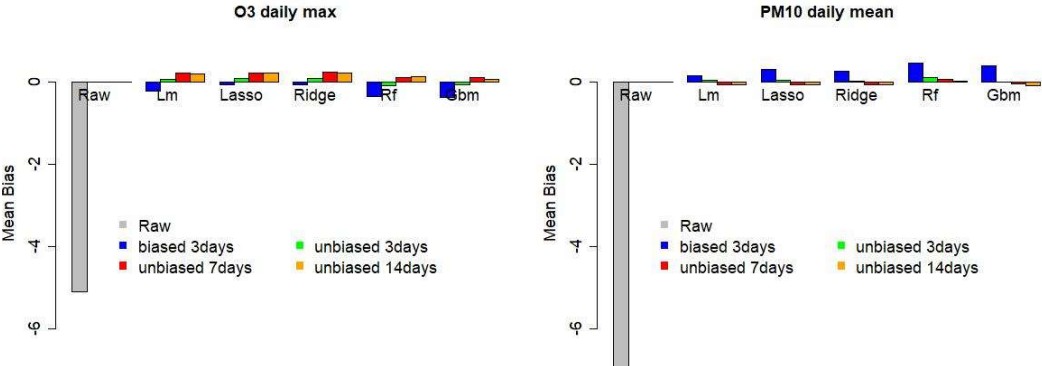


**Figure B3: Mean Bias score for the raw Ensemble model and the global MOS approach with 4 training configurations**

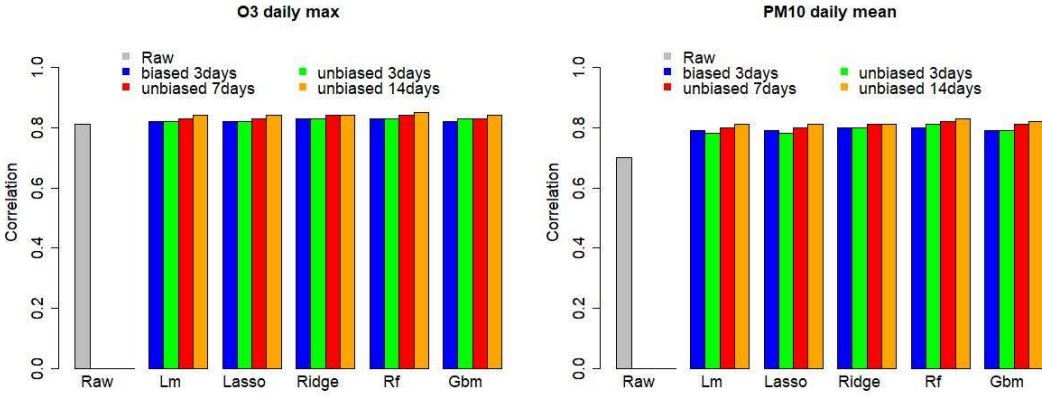

**FigureB4: Correlation score for the raw Ensemble model and the global MOS approach with 4 training configurations**





**Appendix C**





530 **Figure C1: Comparison of the raw Ensemble model and best model scenarios for the local and global MOS approaches. Scores include stations means of RMSE, mean bias and correlation for the prediction of daily mean concentrations over Central Europe (EUC), Northern Europe (EUN), Southern Europe (EUS) and Western Europe (EUW).**

535

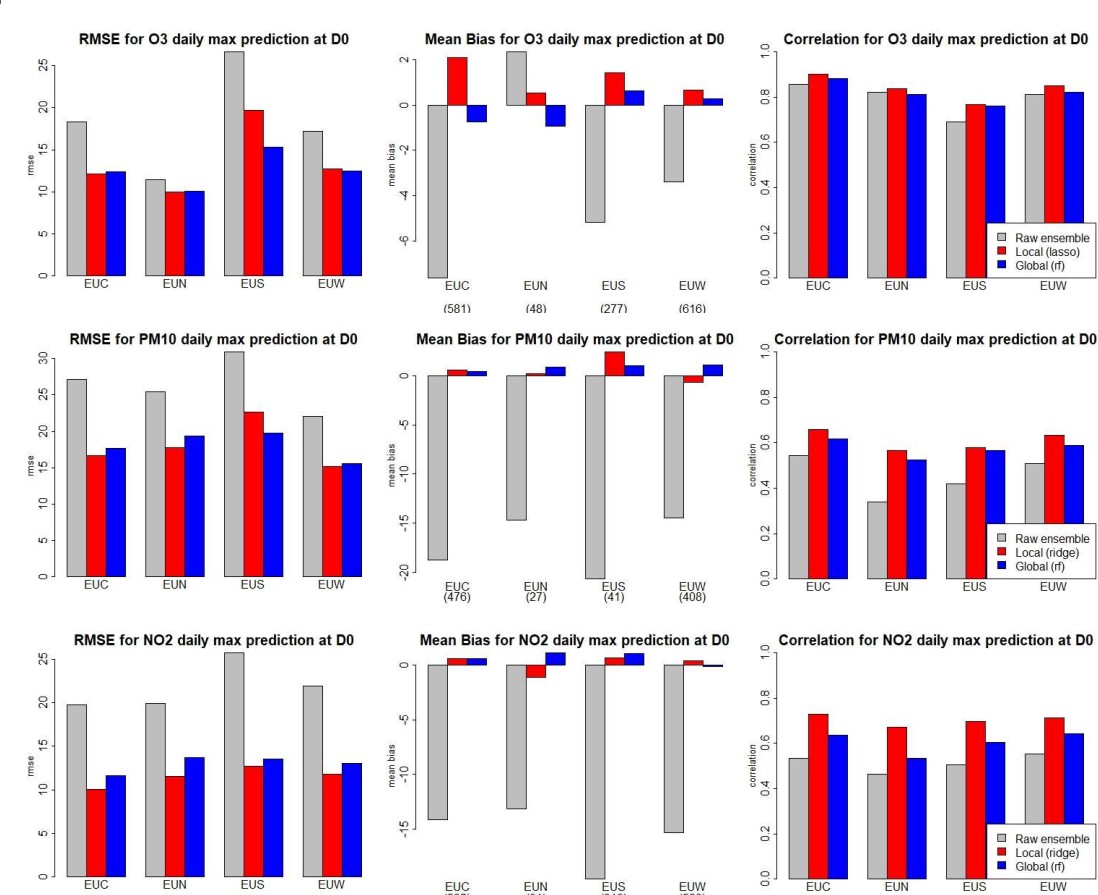





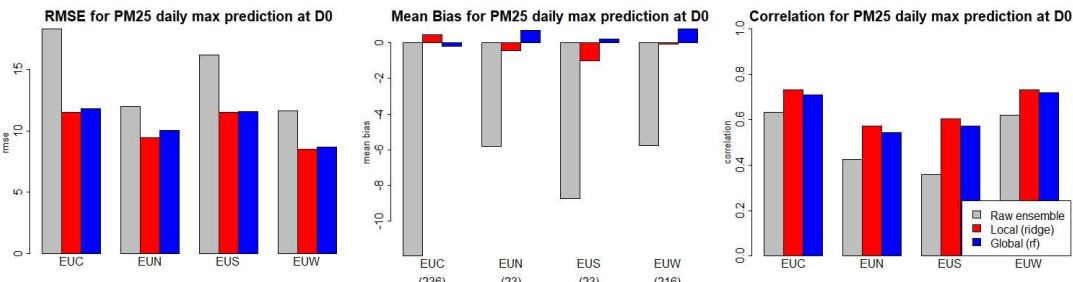

**Figure C2: Comparison of the raw Ensemble model (236) and best model scenarios for the local and global MOS approaches. Scores include stations means of RMSE, mean bias and correlation for the prediction of daily max concentrations over Central Europe (EUC), Northern Europe (EUN), Southern Europe (EUS) and Western Europe (EUW).**

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
