# Peer review of "Technical note: Improving the European air quality forecast of Copernicus Atmosphere Monitoring Service using machine learning techniques"

_Atmospheric Chemistry and Physics, 2022_

## Author Comment (AC1)

This paper investigates the use of Model Output Statistics (MOS) models for correcting the regional CAMS ensemble forecast at the monitoring stations available over Europe for different pollutants (O3, NO2, PM10, PM2.5). Several models (standard and regularized regression, random forest, gradient boosting machine) and approaches (global or station-specific local models, hourly or daily). The postprocessing step is crucial for improving the skills of the AQ forecasts at monitoring stations, and the literature on this topic remains limited. In this context, this study provides a sound and valuable contribution. I found the paper generally clear and well written, although some aspects of their methodology would need to be clarified. As a consequence, I am not entirely sure the authors always follow all the good practices in the field of machine learning, I raised some points that would deserve to be inspected and revised if necessary. I found the quality (resolution) of the figures generally too low, this should be improved. But overall, the study can be accepted for publication in ACP after addressing the comments below.

We would like to thank the referee for its many helpful suggestions. Some aspects have been clarified in the article and the quality of figures has been improved. Our answers for each point are detailed below.

Introduction :

1/

L48-50 : I would not put the "compensation of systematic biases" as another application given that the spatial downscaling intrinsically include some correction of the bias. Also, "compensation" could be replaced by "correction".

Thank you for this comment, the sentence was modified accordingly with this new formulation: "Spatial downscaling is therefore a good example of the relevance of hybrid statistical and deterministic modelling, but correction of systematic biases and better modelling of extreme values can also be achieved at the deterministic model's grid scale."

2/

L59 : I don't think this "hierarchical" term fits with decision tree-based ensemble methods

Thank you. We simply replaced "hierarchical machine learning algorithms" by "tree-based machine learning algorithms".

3/

L63-64 : Some common traditional MOS methods like the aforementioned running-mean or Kalman filter do not require so large input dataset, so I would rephrase this sentence referring to "Some MOS methods…"

The sentence:

"When classical MOS approach is applied at each observation site, long training periods based on model outputs and observations are required."

Was replaced by:

"The classical MOS approach consists in building an individual model at each monitoring site using local data. In this context, some MOS methods (including those based on machine learning algorithms) need long training periods, based on model outputs and observations to reach optimized performances."

4/

L66 : "frequent upgrades" : please specify here the typical frequency of upgrade of CAMS

The sentence: "Every year there are between one and two upgrades in the set-up of the CAMS individual models producing air quality forecasts at regional scale over Europe." was added-up (line 70).

5/

L66-67 : To my opinion, the development of a "global model" including data at numerous monitoring stations will indeed benefit from a larger input dataset for training, but at the same time, it will have to make predictions over a larger range of conditions. Therefore, I don't think there is an intrinsic guarantee of reaching better performance by doing so (although I acknowledge that the authors do get a slightly better performance). Another valuable aspect of the global approach is the ability to correct AQ forecasts at new stations, taking benefit from the other pre-existing ones and without the need to wait for accumulating a sufficiently large local dataset. Eventually, I would try to introduce this idea of the global approach in a more comprehensive way.

There is indeed no guarantee that the global approach gives better performances than the local (with long training period). The first motivation for this alternative global approach is to simplify operational issues such as the maintenance of the MOS to deal with changes in emissions, deterministic model upgrades or the addition of new monitoring sites.

To better introduce the global approach, we added this sentence line 65:

The need for long training period (with constant model formulation over this period) is a difficulty for the maintenance of operational MOS systems since the evolution of pollutants emissions, the upgrades of the deterministic model and the addition of new monitoring sites require frequent re-calibrations of the MOS.

And this sentence line 71:

Therefore, an alternative "global" approach which consists in regularly training a new single model for the whole set of monitoring sites with the most recent data (a few days) has been tested for comparison.

Replaced by

Therefore, an alternative "global" approach, building one single model for all the monitoring sites with a very small training period (a few days preceding the forecast), but using data from the whole geographical domain was also tested for comparison.

Section 2 :

6/

L84 : More specifically, which new SIA were introduced in June 2019 (since I guess nitrates, sulfates and ammonium were already included before) ? As some readers might think from this sentence that no SIA were included before that upgrade, I would be more specific here to avoid confusion.

Indeed no SIA were available in the outputs of CAMS regional forecasts before this upgrade. It was required for this upgrade (June 2019) to provide sia concentrations as the sum of all inorganic species (ammonium, sulfates and nitrates) within PM2.5.

L92 was modified: "and provision of dust (within PM10) and secondary inorganic aerosols (aggregation of ammonium sulphates and nitrates within PM2.5)  in near real time production"

7/

L87-88 : Given that this result is quite counter-intuitive (and somehow contradicts what is explained in the introduction to justify the interest of developing a global approach), I would eventually keep it for latter in the discussion and only mention here that this impact of this upgrade will be discussed later.

 Thank you for this suggestion. Accordingly, the sentence:

"However, the evaluation of the MOS over the 2019 testing period was not strongly impacted by this change in the set-up since the scores remains stable before and after this upgrade"

Has been replaced by:

"The impact of this upgrade on the MOS will be discussed in the conclusion section." (Line 93)

And the following sentence has been added in the conclusion section (line 431):

"As mentioned in section 2, the CAMS ensemble model was subject to un upgrade in June 2019 (i.e. during the testing period). We verified that no break up in the scores occurred during this period and thus consider this upgrade had little impact on the local MOS (despite being calibrated with a slightly different CAMS ensemble version). Nevertheless, we emphasize there is no reason that the local MOS will behave the same way in future upgrades and re-affirm the benefit of the global (short training) MOS approach to deal with those situations."

Section 3 :

8/

L94-96 : This sentence could be better formulated. It corresponds to a supervised learning problem because you have at your disposal a labeled dataset composed of observations of the air pollutant concentrations you are trying to predict.

Thank you, we replaced (line 100):

Since calibration and testing is made possible by the availability of both predictions and observations over a past period, in machine learning terminology it corresponds to a supervised learning problem. A predictive model is built from a training data set that consists in a series of concentration values observed in the past together with the corresponding predictors values. This model fitted with the training data will then be applied to future situations (new predictors values) to produce a statistically corrected concentration forecast.

By

In machine learning terminology it corresponds to a supervised learning problem as we use a training data set composed of a number of predictor variables (also called features) labelled with the corresponding pollutant concentration observations. The model fitted with the training data is then applied to future situations (new predictors values) to forecast pollutant concentrations.

9/

L100 : It would have been interesting to see to investigate the impact of the lead time on the performance of the different methods (although I understand this would have added more results to be discussed and thus less clear take-home messages).

Some tests have been realized at longer forecast lead but they were not exhaustive enough to be integrated in this article.

10/

L104 : Please provide the LASSO acronym when it first appears.

Thank you, the LASSO acronym has been described where it first appears line 109.

11/

L119 : Please provide more information on the difference between ridge and LASSO regression. In contrast with the ridge regression, Tibshirani (1996) notably mentioned that the intrinsic nature of LASSO favors the production of coefficients that are exactly zero (in contrast with ridge regression that produces low non-zero coefficients), which comes with interesting benefits in terms of input features selection (especially when numerous input features) and interpretability.

Thank you, this information has been added line 125:

"In contrast to the ridge regularization, the LASSO tends to produce exactly zero values for those coefficients associated with the less important predictors, offering a way to deal with variable selection and improving model interpretability."

12/

L134 : "reached"

Corrected, thank you.

13/

L141 : The text needs to be revised here. The tuning phase is not done prior to the training phase as it requires the ML model to be trained with different hyper-parameters configurations, in order to select the tuning providing the best performance on the validation dataset. However, when the optimal tuning is obtained, the ML model is trained again with both the training and validation dataset. In a last step, the performance of the obtained tuned ML model can be estimated on a test dataset entirely independent from the previously used training and validation data.

Thank you for this comment, the text has been clarified with this new formulation (line 148):

To reach an optimal balance and optimize the predictive performances, a learning algorithm may be tuned by choosing values for some parameters often referred to as hyper-parameters. The method used for tuning these hyper-parameters consists in a grid search, where possible values for each hyper-parameter are pre-defined. A model is trained and tested for every possible combination of hyper-parameter value using a 5-fold cross validation procedure. The best combination of hyper-parameter is then selected to train the final model, this time using the full training dataset.

14/

L142-144 : For the tuning of the ML models, do I correctly guess that the 5-fold cross-validation is performed exclusively on the training data, i.e. for the local approach only over 2017-2018, always excluding the 2019 test dataset? Please clarify this point in the text. (Obviously, this test dataset should not used in the training/tuning phases as the final evaluation of the performance of the ML models needs to be done on a independent test dataset).

I confirm that the evaluation is performed with an independent testing dataset. Training and testing datasets for both the local and global approaches are described in section 3.2.

15/

L460-471 (appendix A) : "{0, 0.05, 0.1 to 5.0 by increments of 0.1, 6, 7, 8, 10, 12, 15}" I cannot understand what are the values tested here, please explain this point more clearly. Also, please explain the meaning of the different hyper-parameters tested. Finally, please indicate the default value kept for the other hyper-parameters.

Thank you, we propose this new formulation in appendix A to better describe the meaning of each tuning parameter with *caret* R package. We hope that the notation for the grid of values used with the LASSO and ridge models are now unambiguous.

"For both the LASSO and ridge, the penalty coefficient (lambda) is tested with values in {0, 0.05, [0.1 to 5.0 by increments of 0.1], 6, 7, 8, 10, 12, 15}. For the random Forest algorithm, the number of trees (ntree) to grow is fixed to 100 and the number of variables randomly sampled at each split (mtry) is taken as the largest integer less than or equal to the square root of P, where P is the number of predictors. For the GBM algorithm, the number of trees (n.tree) is fixed to 100. The learning rate (shrinkage) takes values in {0.05, 0.1, 0.3}. The number of splits to perform in each tree (interaction.depth) takes values in {2, 7} and the minimum number of observation in a node (n.minobsinnode) takes values in {1, 5}."

16/

L153-159 : Please indicate here that you refer to the MOS approaches that require relatively long input dataset, which is not the case of all MOS methods, as mentioned in the introduction.

Thank you, we added the term "often" (line 164) to indicate that not all the MOS methods are concerned:  A limitation of this local approach (referring to the methods computing a dedicated model per stations) is that it often requires long timeseries of model output and observations (with constant model formulation and set-up over this period) to build an optimized predictive model at each observation site.

17/

L162 : "close"

Corrected, thank you.

18/

L197-199 : Over which period of time these correlations are calculated? Over the training period? Is this changing dynamically? Regarding the global approach, for instance with 3-days training phase, if there are 1000 stations available with full data availability, this means that the correlations are calculated based on 3000 points? Or this is done at each individual station, thus considering only 3 points? Please explain in more details this aspect of your methodology, for both the local and global approach.

Thank you, we added this sentence to clarify this point (L210): "For both the local and global approaches, this correlation is calculated based on the full training dataset, meaning typically with 365 records for a local model built with a 1-year training dataset and 3000 records for a global model based on 1000 monitoring stations spread over the domain and a 3-days training period."

19/

Table 1 : Please clarify if the ML model for O3 only includes O3 raw forecast or also the other pollutant raw forecast in its set of input features (I guess it is the first option). Also, for the hourly approach, please clarify if all the features at the different hours of the day are gathered together without providing any information to the ML model on this hour (or if a specific ML model is trained and used for each individual hour of the day). Please also explain what is done when the observation at D-1 is missing (I guess you simply don't correct the raw forecast ?). If we take the example of a global hourly model with 3-days training set and consider that there are 100 stations available, this means that the training dataset will include 3x100x24=7200 instances, am I understanding correctly?

Thank you for this comment

1- It is actually the second option. We have made it clearer by replacing "It includes the Ensemble forecasts for the 4 pollutants, …" by "It includes the Ensemble forecasts (including the forecasts of the targeted pollutant and the 3 others), …" (line 213).

2- We added this sentence line 206: The same model is used for every hour of the day and the hour of the day is not explicitly passed to the model as a predictor. Considering that the other predictors provide enough information to the model.

3- I confirm, if the observation is not available, the MOS is not computed at the station.

4- yes, I think this as been clarified (see answer to comment 18)

20/

Finally, the use of ridge and LASSO regression requires the input data to be standardized ((X-mean)/std), otherwise the differences of scales of the different predictors can impact in an undesired way the regularization term (lambda). Citing Tibshirani (1997) [The LASSO method for variable selection in the Cox model] : "The lasso method requires initial standardization of the regressors, so that the penalization scheme is fair to all regressors. For categorical regressors, one codes the regressor with dummy variables and then standardizes the dummy variables." Have you applied such standardization? (For RF and GBM, this standardization is not required). If not, I am not sure to which extent this can impact the skills obtained here, but some additional tests would be required to clarify this point.

Thank you for this comment, the glmnet R package used for the fitting of the LASSO and ridge models does perform by default a standardization of the variables prior to the fitting of the model. The categorical « day of week » predictor is indeed coded as a dummy variable prior to standardization.

21/

Just an additional comment but given that the errors of the raw ensemble differ substantially from one region to another (Fig. 3), it might be interesting to allow your global ML model to learn more directly some geographical aspects of the raw ensemble behavior (and errors); one possible approach would be to provide as additional features the climatological meteorological parameters. I am not entirely sure but this might eventually help the ML model to learn specific relationships in European regions of different climate and further improve the skills obtained.

Thank you for this comment, we do think that adding new relevant features could help to improve performances. As a perspective we also mention the possibility to build several global models for distinct European Regions to integrate some geographical aspects (see point 39).

Section 4 :

22/

L229 : Apart from specific MOS approaches (e.g. persistence, running-mean) that are unbiased by construction, it is not correct to say that the MOS models are by construction unbiased, as demonstrated by the results obtained here and shown in Fig. B1. The MOS models used in this study are indeed likely unbiased on the training dataset, but this does not guarantee the absence of bias on the testing dataset (although it is true that if atmospheric conditions are not that different between training and testing datasets and if the test period is sufficiently long, then the bias is expected to be low). Please correct the sentence.

Thank you, we replaced "by construction, the MOS approaches are unbiased and therefore remove a large part of the RMSE"

By "The MOS allows to greatly reduce the bias (see also Appendix B, figure B1) and thus significantly decreases the RMSE." (line 243)

23/

L229-230 : Could you please provide some information regarding the feature importance obtained with RF and GBM models?

Sorry but the analysis of the feature importance was not carried out in the frame of this project.

24/

Figure 1 : The quality of the figures is generally not very good, please increase the resolution. Also, given the small differences and the scale used, please add an horizontal grid on these plots to make it easier to compare.

Figure 1, 2, B1, B2, B3, B4 have been improved with better resolution, larger legend texts and addition of the horizontal grids. We also included the persistence model as suggested in comment number 29. And the following text has been added (line 248):

"In addition to the raw ensemble and the 5 MOS, the persistence model (Pers), a very simple reference model which consists in forecasting for the oncoming day the concentration that was observed at the station during the previous day, is plotted for comparison. Whatever the configuration, the MOS models allow to beat the RMSE score of this persistence model."

25/

Figure 2 : I think it should be "global" rather than "local" in the caption.

Thank you, this has been corrected.

Section 5 :

26/

L269-270 : For the global model, it is not clear why the authors finally selected the one based on only 3 days. Even with several thousands to stations, I would expect a still reasonable computation cost for this global model, no? Could the author provide some information on the computational cost (number of CPUs and hours) ?

In the text "less resource consuming" mainly referred to the local approach for which the CTM might need to be run over long past period to get a training data set with constant model formulation. We chose to present the more realistic scenario (only one year of training data available) for this local approach. For the global approach, we admit the choice can be discussed but we selected the 3-days training scenario because it the one that adapt faster to a change in the modelling system.

This has been clarified line 288:

"For the global approach, we present the 3-days training scenario which is supposed to adapt faster to a change in the modelling system. As mentioned above, performances can be optimized using larger training periods, but we chose to test the scenario which are more prone to cope with operational constraints."

A similar global approach is currently implemented in operational conditions at INERIS using the 3 previous days for training. The training of the 5 MOS models for the 4 pollutant takes approximately 10 minutes using 128 CPU.

27/

Table 2 : Eventually, remove the digit of the relative reduction of RMSE, to make the table easier (lighter) to read (a figure in the Appendix would have been useful to facilitate even more the comparison).

Thank you, we removed the digits as recommended

28/

L324 : "represents"

Corrected, thank you.

29/

L325 : I would also include the persistence method in the previous section. This would provide a very useful reference for judging the results obtained with the different MOS approaches.

The persistence model has been added to figures 1 and 2 (see response to comment number 24).

30/

L328 : "Figs." Corrected, thank you.

31/

L334 : "Figures" Corrected, thank you

32/

L340 : "allows" Corrected, thank you

33/

L342 : Introduce the acronym before

Done line 339, thank you.

34/

Figures 5-8 : Again the quality of these figures is very low, please increase the resolution, the size of the labels as well as the size of the legend (grey circle and square cannot be distinguished in this legend panel). Also, you could regroup them into two 4-panels figures, one for O3, the other for PM, to facilitate comparison between daily and hourly results.

Thank you, the resolution, size of labels and legends have been increased figures 4, 5 and 6.

Figures 5 to 8 have been grouped into two 4-panels figures as recommended (figures 5 and 6) and the text has been reorganized accordingly (lines 349 to 380).

35/

Pages 14-15 : Please organize your text in several paragraphs.

Thank you, the text is now organized in several paragraphs.

36/

L349 : Please provide some numbers of the duration of the episodes in your domain and period of study. Also, in this section, please indicate the frequency of observation of these O3 and PM exceedances (base rate), this is an important piece of information that is currently missing.

This information has been added in the text line 338:

"This exceedances events remain relatively rare. In our 2019 testing dataset (only summer months for $O_3$), the base rate is 1.3% and 2% respectively for $O_3$ and $PM_{10}$ exceedances. In average, the duration of these episodes of exceedances at a station is 1.6 days for $O_3$, with 30% of the episodes lasting 2 days or more and 4% lasting 5 days or more. For PM10, the episodes tend to be a little bit larger, with an average duration of 1.8 days, 40% of the episodes lasting 2 days or more and 5% lasting 5 days or more."

37/

L358-361 : To further demonstrate your point, could you provide the scores obtained over these 34% stations and over the remaining 66%? (same for PM)

Thank you for that suggestion. Actually, this point was not correct or at least it does not explain entirely why the random forest and GBM does not work as well as the linear models for the detection of O3 exceedances (considering the local approach). As suggested, we plotted the scores (see the performance diagram below) for the subset of station which had been exposed to exceedances during the training period and the results were quite similar to those for the whole set of station, with linear models scores still significantly better than random forest and GBM scores. Further analysis is needed to understand this phenomenon and it will not be explained in the frame of this article. We therefore removed the part of the text mentioning the incapacity of tree-based model to extrapolate outside the range of the training data as a possible explanation.

[Figure]

38/

L410-411 : I think there might be an error here. The global/hourly GBM approach gives better results on PM10 (Fig. 8) than O3 (Fig. 7). Please clarify.

The sentence was ambiguously formulated. What we meant to say is that GBM is more adapted *than random forest* for the detection of O3 exceedances. This has been clarified line 416:

However, if the MOS is designed for hourly prediction, the Gradient Boosting Machine (GBM) algorithm is more adapted *than random forest* to detect $O_3$ daily max threshold exceedances.

Conclusion :

39/

Based on all these results and acknowledging the persistent limitations that remain to be overcome, could the authors make any personal recommendation on which approach would be the best for an operational implementation in CAMS ? Based on the experience acquired, could the authors provide some perspectives to be explored in order to further improve the skills obtained here?

For a CAMS implementation, we would recommend the use of the global MOS approach as it allows to adapt automatically and in near real time to any upgrade in the CAMS ensemble model. We also mention in the conclusion (and this should be confirmed with future studies) that this approach could better adapt to rapid changes in emission sources (we mentioned the example of the reduction of emissions due to the COVID-19 crisis). The choice for the best algorithm depends on the scores one wants to optimize. For an hourly forecast production, we would recommend the GBM model which has the advantage of producing the most balanced detection skills (with reasonable probability of detection and success ratio scores) for the O3 pollutant (while the other models have very low probability of detection for this pollutant). For this global approach we would also recommend a 3 to 7 days training period, offering good performances within reasonable computing time and also having in mind that the shorter the training period, the faster the MOS adapt to upgrades of the modelling system.

Although a result of this study is that adding up new predictors to the MOS does not necessarily improve the performances, we believe there is still ground for improvements finding new relevant set of predictors to feed the MOS. Another perspective to be explored considering the global approach is the possible gain obtained by multiplying the number of global models for example, defining one global model per macro-region, per station typology etc… Lastly, performances could certainly be optimized, especially for the GBM model by exploring more exhaustively the possible tuning values for hyper-parameters.

---

## Author Comment (AC2)

The authors propose a treatment to an ensemble of CTMs to improve air quality forecasting. This has been done repeatedly in the past using various approaches. The novelty here is the systematic use of ML methods which seems to produce promising results, that is, outscoring the current ensemble method. As I am not a ML expert, I cannot judge the technical implementation of the various algorithms tested. Overall though, I would say the analysis presented makes sense and I trust the authors that the treatments they propose 'are doing the right thing for the right reason'.

My advice to the editor is to accept the manuscript for publication, pending some clarifications that I invite the authors to consider:

We would like to thank the referee for its positive comments and helpful suggestions. Our answers for each point are detailed below.

- Quality of figure needs improving

  The quality of all figures has been improved, with better resolution and increased size of legends and labels.

- Specify in plain words what is meant by raw ensemble – is that the unbiased ensemble mean? Possibly I have overlooked, but I cannot locate a definition in the text

  Thank you, this has been clarified line 218: For the global approach, tests have been performed using as a predictor either the raw Ensemble (i.e. the median of the 7 individual deterministic models) forecast or the unbiased Ensemble concentration of the target pollutant. The unbiased concentration is defined as the forecasted Ensemble concentration minus the bias observed at the station during the previous days (days of the chosen training period).

- Please comment on what would make your ML methodology better/preferred to other ensemble-improving methods (as for example: https://acp.copernicus.org/articles/14/11791/2014/, https://acp.copernicus.org/articles/13/7153/2013/)

  We would not say that our MOS methodology is better than the ensemble methods you mention but rather that they are complementary. The MOS could be applied to any type of ensemble model to downscale gridded concentration forecasts and further improve the performances at the locations of monitoring sites. Of course, the closer the ensemble is to the observations, the harder it will be to improve the performances with the MOS. Also, note that the MOS methodology we present was applied to the CAMS ensemble outputs but it could also be applied to a single deterministic model's outputs. This might be an advantage in situations where ensemble of several models is not available.

- I believe the authors could make stronger conclusions had they tested their methodology on high pollution episodes, which are notoriously more difficult to predict.

  We did test the ability of the methodology to detect high pollution episodes (exceedances of the European regulatory threshold values) for ozone and PM10

pollutants. This evaluation is made using the performance diagram which synthesize 4 detection scores in section 5

- On the same line of the comment above, would the use of the proposed ML method improve on the predicting of exceedances for regulated pollutant? Please consider adding a comment on these.

  Prediction of threshold exceedances has been evaluated (see response for the previous point) and results are mentioned in the conclusion section. Indeed, the use of a MOS method usually improves the ability to detect threshold exceedances.

- What do you think are the implication of your proposed methodology on gridded output?

  The MOS methodology we propose is designed to correct/improve concentration forecasts at the locations of monitoring sites. Additional post-processing needs to be applied to the MOS forecasts in order to spatialize the output concentrations at any grid cells of the modelling domain. Such post-processing could be based on land use regressions or kriging technics as in the French national forecasting system PREV'AIR (Honoré et al., 2008; Rouïl et al., 2009).

- Please avoid the use of acronyms in the conclusion*

  Thank you, CSI (Critical Success Index) acronym has been suppressed in the conclusion, referring in a more general way to detection performances. CAMS, MOS and GBM acronyms are now re-introduced in the conclusion section.